# Prevalence and Predictors of the Use of Low-Calorie Sweeteners Among Non-Pregnant, Non-Lactating Women of Reproductive Age in Australia

**DOI:** 10.3390/nu16223963

**Published:** 2024-11-20

**Authors:** Mumtaz Begum, Shao-Jia Zhou, Saima Shaukat Ali, Zohra S. Lassi

**Affiliations:** 1Adelaide Medical School, Faculty of Health and Medical Sciences, The University of Adelaide, Adelaide, SA 5005, Australia; mumtaz.begum@adelaide.edu.au; 2Robinson Research Institute, The University of Adelaide, Adelaide, SA 5000, Australia; jo.zhou@adelaide.edu.au (S.-J.Z.); saima.shaukatali@adelaide.edu.au (S.S.A.); 3Department of Food and Nutrition, School of Agriculture, Food and Wine, The University of Adelaide, Adelaide, SA 5000, Australia; 4School of Public Health, Faculty of Health and Medical Sciences, The University of Adelaide, 50 Rundle Mall Plaza, Rundle Mall, Adelaide, SA 5000, Australia

**Keywords:** artificial sweetener, non-nutritive sweetener, women of reproductive age, latent class analysis, Australia

## Abstract

Objective: There is concern about the potential health implications of low-calorie sweetener (LCS) consumption. This study aimed to determine the prevalence and patterns of LCS use among women of reproductive age (WRA) in Australia. Methods: This cross-sectional study involved a two-stage analysis. First, latent class analyses (LCA) were employed to identify patterns of LCS use. Subsequently, regression analyses were conducted to assess the association between sociodemographic and lifestyle characteristics and the two outcomes: (1) self-reported LCS use, and (2) the identified LCS consumption patterns/classes. Results: A total of 405 WRA completed the survey (mean age 32.0 ± 8.6 years, mean BMI 28.71 ± 11.1 kg/m^2^)_,_ with 44.7% reporting LCS consumption. LCA analysis identified three distinct LCS consumption patterns: light users (45.9%), moderate users (26.0%) and heavy users (28.6%). A high proportion of participants did not meet the Australian dietary guidelines for recommended servings of vegetables (57.8%), dairy (44.2%), meat (48.2%) and grains (74.8%). Compared to Caucasian women, those from South Asian backgrounds (OR 4.16; 95% CI 1.71–10.1) and Aboriginal and Torres Strait Islander women (OR 1.40; 95% CI 0.42–4.63) were more likely to use LCS. Women who participated in the weight loss programs, with overweight/obesity, and those using LCS for weight loss purposes were more likely to be moderate or heavy LCS users than light users. Additionally, socioeconomically disadvantaged women were less likely to be moderate or heavy LCS users. Conclusions: This study highlights the widespread use of LCS among WRA in Australia, with distinct consumption patterns influenced by cultural, health-related, and socioeconomic factors. These findings underscore the need for targeted interventions to promote healthy eating practices within this population.

## 1. Introduction

The global food and beverage industry has increasingly shifted towards healthier options, driven by a growing awareness of the link between diet and health [1]. A significant trend within this shift is the widespread use of artificial sweeteners [1], also known as non-nutritive sweeteners or low-calorie sweeteners (LCSs), in various products. These sweeteners are marketed as healthier alternatives to sugar, particularly for weight management, and have gained popularity for providing sweetness without the calories of traditional sucrose. However, recent evidence has challenged the purported benefits of LCSs, with some studies suggesting that LCS consumption may be associated with adverse health effects, including obesity [2], diabetes [3,4,5,6,7], cardiovascular problems [3,4,5,6,7], depression [8], dementia [9], osteoporosis [10], and some cancers [11]. Despite these findings, the evidence remains inconsistent [12,13].

The Dietary Guidelines for Americans (2015) indicate that LCS may be useful for short-term weight management, but the long-term effectiveness of LCS remains unclear [14]. In 2023, the World Health Organization (WHO) released new guidelines advising against the use of LCSs for weight control, citing potential undesirable health implications from long-term consumption [15].

As LCSs become more prevalent, understanding their usage and health effects becomes increasingly important. The impact of LCSs on non-pregnant and non-lactating women of reproductive age (WRA) is of particular concern due to the potential long-term effects on their health and the health of future generations [16]. Current research tends to focus on the general population [2,3,4,5,6,7] and often overlooks the specific dietary needs and potential health consequences for women in their reproductive years. Furthermore, existing studies frequently aggregate different age groups, making it challenging to identify consumption patterns specific to this life stage.

The absence of comprehensive data on LCS intake among non-pregnant and non-lactating WRA impedes a thorough understanding of how these dietary choices might influence metabolic health, hormonal balance, and overall well-being. Insights into the prevalence and usage patterns of LCS among this group are essential for developing targeted health interventions and dietary recommendations tailored to their unique needs. Such knowledge can also inform future research and policy efforts aimed at improving health outcomes for non-pregnant and non-lactating WRA and their families.

The objective of this study was to determine the prevalence and predictors of LCS consumption among WRA living in Australia.

## 2. Methods

In this cross-sectional study, we collected survey data from WRA (18–49 years) living in Australia who were not currently pregnant or breastfeeding, had consented to participate, and had a good understanding of the English language. This web-based survey was developed and distributed via an anonymous link using Qualtrics, a reputable and widely used online survey tool that allows researchers to create, test, and distribute surveys online. Screening questions at the beginning of the survey ensured that the participants met the inclusion criteria, including age, gender, pregnancy or breastfeeding status, and residence in Australia.

The prevalence of LCS consumption among WRA is reported to range between 30% and 45% [17]. Therefore, an average prevalence of 40% was used for the sample size estimation, with 90% precision and 80% power, requiring a minimum of 369 participants. We collected data from 405 non-pregnant, non-lactating WRA.

The questionnaire consisted of sociodemographic questions, fourteen questions regarding LCS consumption and the BEVQ-15 (a validated beverage intake questionnaire) [18]. A short six-item dietary screening tool (Australian Short Dietary Screener), which has been validated in an older population [19,20], was used to assess the intake of the Five Food Groups (vegetables, fruit, meat and alternatives, dairy, and grains) as defined by the Australian Dietary Guidelines (ADG) [21]. The short six-item tool was used to assess compliance with Australian dietary guidelines by asking about the estimated number of servings consumed during an average week from each of the five food groups defined in the Australian Dietary Guidelines. Respondents were asked to report the number of servings they eat from each food group per day or during an average week. Examples of one serving for each food group were provided in the questionnaire (based on Australian Dietary Guidelines descriptions).

Sociodemographic information included in the questionnaire included age, education, employment status, state of residence, postcode, and country of birth. Lifestyle-related questions covered exercise habits and reasons for using LCS. LCS use in our paper included the use of any non-nutritive sweeteners, low-energy sweeteners, artificial sweeteners, and sugar substitutes, which are usually low in calories or have no calories.

Using the Beverage Questionnaire (BEVQ-15) [18], we asked respondents to report the frequency of consumption of these beverages over the past month using a six-point scale, ranked as follows: never, <1 time per week, once per week, 2–3 times per week, 4–6 times per week, and daily. Additionally, we inquired whether respondents regularly consume other LCS-containing products, such as jam/jellies, candies, and condiments.

## 3. Outcome

The outcome of the study regarded the prevalence and patterns of LCS use, and their association with sociodemographic characteristics.

## 4. Covariates

The sociodemographic factors included in our study were age, marital status, ethnicity and country of birth, and some socioeconomic measures included employment status, educational attainment, and housing condition. Additional covariates comprised the Body Mass Index (BMI) categories (<18, 25–29, 30–34, ≥35 kg/m^2^), overall health status (excellent, fair, good, poor), presence of any medical condition (yes/no), participation in a weight loss program (yes/no), reasons of LCS use (professional advice, for diabetes management, for taste or other reasons), dietary restrictions, and diet quality.

To assess the level of moderate physical activity, we asked how often they participate in moderately intense exercise (defined as any activity that requires some effort but the person is able to maintain conversation while performing it). Responses were categorized into none, 1–3 times per week, 4–6 times/week, and daily.

We assessed compliance with Australian dietary requirements using a six-item tool (Australian Short Dietary Screener [19,20]), wherein participants were asked to report the number of servings consumed per day from each food group during an average week. To aid their responses, examples with pictures illustrating serving sizes for each food group were provided.

Responses were dichotomized into two categories: (1) meeting the recommended number of servings for each food group (i.e., adhering to the Australian dietary guidelines) or (2) consuming fewer than the recommended number of servings for each food group (i.e., not meeting the dietary guideline). We then combined these dichotomized responses into six categories; (1) did not meet the dietary guideline of all five food group, (2) met the dietary guidelines of one food group, (3) met the dietary guidelines of two food groups, (4) met the dietary guidelines of three food groups, (5) met the dietary guidelines of four food groups, and (6) met the dietary guidelines of all five food groups.

## 5. Statistical Analyses

The analyses were performed in two stages: (1) identification of LCSs use patterns through latent class analyses (LCA), and (2) assessment of associations between sociodemographic and lifestyle characteristics and the two outcomes (1) reported LCSs use (2) the identified patterns of LCSs use.

Analyses were conducted using Stata (MP 17.0).

## 6. Latent Class Analysis

In the first phase of analyses, we conducted the LCA to identify the pattern of LCS consumption. LCA is a robust statistical method used to uncover subgroups of individuals (i.e., latent classes) within a population based on patterns of responses to observed categorical variables. For this analysis, we included nine observed categorical variables in the latent class model: use of soda, energy drinks, sports drinks; coffee, black tea or milk tea; yogurt and milk, jam/jellies, candies, and condiments. For the LCS model, we used intake of soft drinks (soda, energy drinks, sports drinks) and coffee (black tea or milk tea); yogurt and milk were categorized into four categories (never or <1 per week, once a week, 2–3 times per week, ≥4 times per week/daily). The regular use of the rest of the variables included in LCS model was coded as binary.

We selected a three-class model based on the lowest Akaike Information Criterion (AIC), lowest Bayesian Information Criterion (BIC), and high entropy values, which indicated a good model fit. The three identified classes were: Class 1 (light LCS users), Class 2 (moderate LCS users), and Class three (heavy LCS users) (Appendix A).

In the second phase of analyses, we used logistic regression and multinomial logistic regression to assess the association between sociodemographic characteristics and the two outcomes, (1) reported LCS use and (2) identified pattern of LCS use, respectively. The analyses were adjusted for all the covariates mentioned in the methods section.

## 7. Ethics Approval

Ethics approval for this study was obtained from the Human Research Ethics Committee of the University of Adelaide (H-2023-147).

## 8. Results

Overall, 405 WRA completed the survey, with a mean age of 32.0 ± 8.6 years, and a mean BMI of 28.71 ± 11.1 kg/m^2^. About three-quarters (75%) of the women were employed, 40% had completed a bachelor/higher degree, 71% were Caucasian, and 4.2% were Aboriginal or Torres Strait Islander women. These characteristics are reflective of the broader demographics of WRAs in Australia (Table 1).

Among the 405 women who responded to our survey, 44.7% reported consuming LCSs. The LCA of LCSs consumption across various foods and beverages identified three distinct usage patterns: 45.9% of participants were classified as light users (LCS class 1), 26.0% as moderate users (LCS class 2), and 28.6% were classified as heavy LCS users (LCS Class 3).

Table 2 presents the characteristics of women based on their LCSs usage. Among those who reported consuming LCSs, a higher proportion were of South Asian origin or were of Aboriginal or Torres Strait Islander descent, identified as vegetarians, participated in weight loss programs, and used LCSs for weight management. Among heavy LCS users, a high proportion were of South Asian origin, followed dietary restrictions (such as Halal or vegetarian diets), met the Australian dietary guidelines, and were younger (18–25 years old).

Appendix A illustrates the probability of consuming LCSs from different foods and beverages for each latent class. Around 45% of the sample were classified as light LCS users (Class 1), based on their probabilities of LCS consumption from various food items. In this group, a high proportion of women rarely or never consumed LCSs (never or <1 per week) from black tea (85%), white tea (84%), soft drinks (83%), and yoghurt/milk (72%).

Approximately 26% of the sample was categorized into class two (moderate users). A large proportion of them also rarely or never consumed LCSs (never or <1 per week) from black tea (80%), white tea (79%), and yogurt/milk (60%). However, they had a higher frequency of LCS consumption (more than or equal to four times/week to daily) from soft drinks (34%), LCS-containing soda or any low-calorie sweeteners (Appendix A).

Approximately 28% of the women were classified into class three (labeled as heavy users). A high proportion of this group reported using LCS almost daily (more than or equal to four times/week or daily) from black tea (45%), white tea (46%), soft drinks (33%), and yogurt/milk (44%) (Appendix A).

Figure 1 shows the proportion of women who consumed the recommended number of servings for each food group, following the Australian dietary guidelines. A high proportion of women did not consume the recommended servings of vegetables (57.8%), dairy (44.2%), meat (48.2%) and grains (74.8%).

Table 3 displays the sociodemographic characteristics associated with reported LCS use. Compared with younger women (18–25 years), those aged 36–49 were less likely to use LCSs (OR 0.42; 95%CI 0.21, 0.84). Additionally, compared with Caucasian women, women from South Asian backgrounds (OR 4.16; 95%CI 1.71, 10.1), and Aboriginal and Torres Strait Islander women (OR 1.40; 95% CI 0.42, 4.63) were more likely to use LCSs. Women who participated in weight loss programs (OR 1.92; 95% CI, 1.05, 3.51) had higher odds of using LCSs compared to women who never participated. Moreover, women who did not meet the dietary guidelines had a higher likelihood of using LCSs compared to those who met the guidelines.

Table 4 presents the sociodemographic characteristics associated with different patterns of LCS use (light, moderate, and heavy use). Women who participated in the weight loss program, who had high BMIs, and who were using LCSs for weight loss reasons were more likely to be moderate or heavy LCS users than light users. In addition, socioeconomically disadvantaged women (those with lower education, who were divorced/widowed, or who were retired or unable to work) were less likely to be moderate or heavy LCS users.

## 9. Discussion

The findings from this Australia-wide survey provide valuable insights into the prevalence, patterns, and correlates of LCS consumption among non-pregnant and non-lactating WRAs.

The prevalence of reported LCS use in our sample was 44.7%, indicating a significant proportion of women consuming LCSs. Our LCA identified distinct patterns of LCS use within the sample. Specifically, nearly half of the participants were classified as light users, a quarter as moderate users, and the remaining as heavy users. These findings highlight the heterogeneous nature of LCS consumption habits among women, emphasizing the importance of considering individual variability when assessing dietary patterns. Understanding these distinct patterns can help tailor public health messages and interventions more effectively.

Notably, a substantial proportion of women did not meet the recommended dietary guidelines for key food groups, indicating potential gaps in dietary quality. The Australian Longitudinal Study on Women’s Health (ALSHW) reported that only 2%, 10%, 22%, and 10% of women from all age groups meet the daily recommended intakes of vegetables, grains, dairy, and meat and meat substitutes, respectively [22]. This underscores the need for targeted interventions to promote healthier dietary habits among non-pregnant and non-lactating women of reproductive age. The gap between actual dietary practices and recommended guidelines highlights an area for improvement in public health nutrition strategies.

Our analysis identified several sociodemographic factors associated with LCS use. Women from South Asian backgrounds, Aboriginal and Torres Strait Islander women, those participating in weight loss programs, and those not meeting dietary guidelines were more likely to use LCSs. Notably, LCS consumption is reported to be higher among the Aboriginal and Torres Strait Islanders populations compared to their non-indigenous counterparts [23]; the high prevalence among culturally and linguistically diverse (CALD) communities, particularly women of South Asian origin, is intriguing and warrants further exploration. Given that individuals from South Asian backgrounds have a significantly higher risk of developing diabetes compared to other population groups in developed countries [24], addressing these factors in public health interventions could help reduce LCS intake and promote healthier eating habits.

Further, our study revealed distinct patterns of LCS usage based on sociodemographic characteristics. Women participating in weight loss programs, those with higher BMI, and those using LCS for weight loss purposes were more likely to be moderate or heavy users. These findings highlight the importance of preventive public health measures to promote healthier weight management strategies, rather than relying on LCSs [14], especially since higher LCS consumption has been linked with obesity [3].

The study has several strengths and limitations. One significant strength is that it is the first nationwide survey in Australia to determine the use of LCSs among non-pregnant and non-lactating WRAs, providing valuable baseline data for future research and public health initiatives. However, there are limitations. First, as a cross-sectional study, it captures data at a single point in time, limiting our ability to infer causality or observe trends over time. Therefore, we cannot determine whether LCS consumption is increasing or decreasing, or how it might relate to long-term health outcomes. Second, the underrepresentation of CALD and Indigenous populations in the sample means that the results may not fully reflect the dietary habits and LCS consumption patterns of these groups. There could be measurement bias in our assessment of compliance with Australian dietary guidelines, as we used a short six-item tool [20], which does not capture the detailed intake of individual food items within each food group, although it has been validated in older age groups. Finally, class assignment as part of latent class analysis is based on probabilities, therefore exact class assignment is not guaranteed.

Overall, our findings contribute to a better understanding of LCS consumption among WRAs, highlighting the need for targeted interventions to promote healthier dietary habits and reduce excessive LCS intake in this population. Future research should explore longitudinal trends in LCS consumption and the impact of interventions aimed at promoting healthier dietary choices. Additionally, understanding the long-term health outcomes associated with varying levels of LCS consumption can inform more effective public health policies and dietary guidelines. Given the high prevalence of LCS use observed in this study, further research is strongly recommended, particularly with a focus on underrepresented groups such as CALD and indigenous populations, to provide a more comprehensive understanding of LCS consumption across all segments of the population. Longitudinal studies are also needed to track changes in LCS use over time and to assess the long-term health impacts of such dietary habits. This continued research is essential for informing targeted public health interventions and promoting healthier dietary practices among WRAs in Australia.

## 10. Conclusions

In conclusion, this Australia-wide survey offers valuable insights into the prevalence, patterns, and sociodemographic correlates of LCS consumption among WRAs. The study highlights the widespread use of LCSs, especially among women from south Asian backgrounds, Aboriginal and Torres Strait Islander women, those participating in weight loss programs, and those not meeting dietary guidelines. In addition, many women did not meet recommended dietary guidelines, indicating a need for improved dietary quality. These findings underscore the importance of targeted public health interventions to promote healthier eating habits and reduce excessive LCS intake. Future research should focus on longitudinal trends in LCS consumption and evaluate the effectiveness of interventions designed to foster healthier dietary choices in this population, ultimately contributing to better health outcomes for WRAs.

## Figures and Tables

**Figure 1 nutrients-16-03963-f001:**
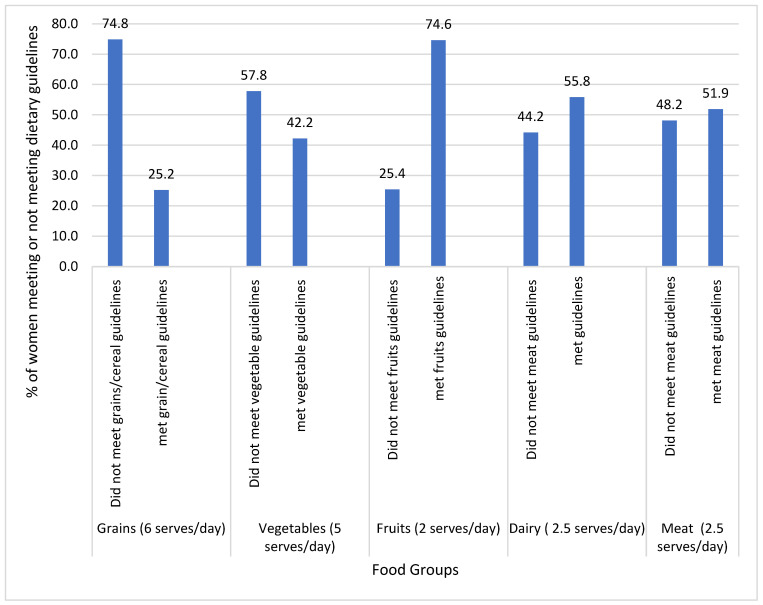
Proportion of women who met the Australian dietary guidelines (N = 405).

**Table 1 nutrients-16-03963-t001:** Characteristics of reproductive age women in Australia (survey participants).

	N = 405
Characteristics	*n* (%)
Do you consume any low-calorie sweeteners (LCSs)?	
No	224 (55.3)
Yes	181 (44.7)
Age categories	
18–25 years	112 (27.7)
26–35 years	142 (35.1)
36–49 years	151 (37.3)
Ethnicity	
Aboriginal/Torres Strait Islander	17 (4.2)
Asian/Pacific Islander	30 (7.4)
Other	25 (6.2)
South Asian	44 (10.9)
White/Caucasian	289 (71.4)
Are you currently employed?	
A homemaker	59 (14.6)
Employed	302 (74.6)
Early Retirement, unable to work	44 (10.9)
Education	
Bachelors and above	162 (40.0)
Certificate or Diploma	124 (30.6)
Completed year 12	96 (23.7)
others	23 (5.7)
Marital Status	
Married or de facto relationship	192 (47.4)
Single/never married	188 (46.4)
Divorced/Widowed	25 (6.2)
Country of birth	
Australia	331 (81.7)
Other	74 (18.3)
Housing	
Other	39 (9.6)
Own the property outright	43 (10.6)
Own with mortgage	115 (28.4)
Rent	208 (51.4)
Body mass Index (BMI)	
<18	15 (3.7)
25–29	182 (44.9)
30–34	76 (18.8)
≥35	132 (32.6)
Health status	
Excellent	46 (11.4)
Fair	92 (22.7)
Good	255 (63.0)
Poor	12 (3.0)
Ever Participation in weight program?	
No	329 (81.2)
Yes	76 (18.8)
Reasons of LCS use	
Health professional advice	30 (7.4)
Diabetes management	29 (7.2)
weight loss	175 (43.2)
Other	50 (12.3)
Taste better	121 (29.9)
Dietary restrictions	
Do not follow any dietary restrictions	313 (77.3)
Food allergies/lactose intolerance etc	40 (9.9)
Halal	13 (3.2)
Vegetarian	25 (6.2)
Others	14 (3.5)
Meeting Australian dietary guidelines	
Did not meet dietary guidelines if any food group	54 (13.3)
Met dietary guidelines of one food group	101 (24.9)
Met dietary guidelines of two food groups	60 (14.8)
Met dietary guidelines of three food groups	49 (12.1)
Met dietary guidelines of four food groups	62 (15.3)
Met dietary guidelines of all five food groups	79 (19.5)
Exercise	
None	83 (20.5)
1–3 times per week	244 (60.2)
4–6 times per week	53 (13.1)
Every day	25 (6.2)
Medical condition	
No	267 (65.9)
Yes	138 (34.1)

**Table 2 nutrients-16-03963-t002:** Characteristics of women of reproductive age by low-calorie sweeteners (LCSs) use.

	Do You Use LCS?		The Pattern of LCS Use from Latent Class Analyses		
	No	Yes	Class 1 Light LCS Use	Class 2 Moderate LCs Use	Class 3 Heavy LCS Use
N	224	181	186	103	116
	*n* (%)	*n* (%)	*n* (%)	*n* (%)	*n* (%)
Age					
18–25 years	52 (23.2)	60 (33.1)	46 (24.7)	23 (22.3)	43 (37.1)
26–35 years	75 (33.5)	67 (37.0)	60 (32.3)	45 (43.7)	37 (31.9)
36–49 years	97 (43.3)	54 (29.8)	80 (43.0)	35 (34.0)	36 (31.0)
Ethnicity					
Aboriginal/Torres Strait Islander	7 (3.1)	10 (5.5)	7 (3.8)	3 (2.9)	7 (6.0)
Asian/Pacific Islander	22 (9.8)	8 (4.4)	17 (9.1)	3 (2.9)	10 (8.6)
Other	15 (6.7)	10 (5.5)	12 (6.5)	7 (6.8)	6 (5.2)
South Asian	14 (6.2)	30 (16.6)	17 (9.1)	10 (9.7)	17 (14.7)
White/Caucasians	166 (74.1)	123 (68.0)	133 (71.5)	80 (77.7)	76 (65.5)
Employment status					
Homemaker	38 (17.0)	21 (11.6)	26 (14.0)	16 (15.5)	17 (14.7)
Employed	162 (72.3)	140 (77.3)	132 (71.0)	81 (78.6)	89 (76.7)
Early Retirement, unable to work	24 (10.7)	20 (11.0)	28 (15.1)	6 (5.8)	10 (8.6)
Education					
Bachelors and above	88 (39.3)	74 (40.9)	65 (34.9)	47 (45.6)	50 (43.1)
Certificate or Diploma	68 (30.4)	56 (30.9)	63 (33.9)	30 (29.1)	31 (26.7)
Completed year 12	53 (23.7)	43 (23.8)	43 (23.1)	23 (22.3)	30 (25.9)
Others	15 (6.7)	8 (4.4)	15 (8.1)	3 (2.9)	5 (4.3)
Marital Status					
Married or de facto relationship	103 (46.0)	89 (49.2)	82 (44.1)	52 (50.5)	58 (50.0)
Single/never married	103 (46.0)	85 (47.0)	89 (47.8)	46 (44.7)	53 (45.7)
Divorced/Widowed	18 (8.0)	7 (3.9)	15 (8.1)	5 (4.9)	5 (4.3)
Country of birth					
Australia	182 (81.2)	149 (82.3)	148 (79.6)	88 (85.4)	95 (81.9)
Other	42 (18.8)	32 (17.7)	38 (20.4)	15 (14.6)	21 (18.1)
Housing					
Other	20 (8.9)	19 (10.5)	20 (10.8)	10 (9.7)	9 (7.8)
Own the property outright	24 (10.7)	19 (10.5)	17 (9.1)	8 (7.8)	18 (15.5)
Own with mortgage	68 (30.4)	47 (26.0)	52 (28.0)	34 (33.0)	29 (25.0)
Rent	112 (50.0)	96 (53.0)	97 (52.2)	51 (49.5)	60 (51.7)
Body mass Index (kg/m^2^)					
<18	5 (2.2)	10 (5.5)	8 (4.3)	2 (1.9)	5 (4.3)
25–29	109 (48.7)	73 (40.3)	94 (50.5)	34 (33.0)	54 (46.6)
30–34	37 (16.5)	39 (21.5)	31 (16.7)	26 (25.2)	19 (16.4)
≥35	73 (32.6)	59 (32.6)	53 (28.5)	41 (39.8)	38 (32.8)
Health Status					
Excellent	26 (11.6)	20 (11.0)	21 (11.3)	10 (9.7)	15 (12.9)
Fair	45 (20.1)	47 (26.0)	40 (21.5)	24 (23.3)	28 (24.1)
Good	145 (64.7)	110 (60.8)	119 (64.0)	67 (65.0)	69 (59.5)
Poor	8 (3.6)	4 (2.2)	6 (3.2)	2 (1.9)	4 (3.4)
Ever Participation in weight program					
No	192 (85.7)	137 (75.7)	169 (90.9)	75 (72.8)	85 (73.3)
Yes	32 (14.3)	44 (24.3)	17 (9.1)	28 (27.2)	31 (26.7)
Reasons for LCS use					
Health professional advice	24 (10.7)	6 (3.3)	23 (12.4)	3 (2.9)	4 (3.4)
Diabetes management	20 (8.9)	9 (5.0)	16 (8.6)	5 (4.9)	8 (6.9)
weight loss	84 (37.5)	91 (50.3)	65 (34.9)	59 (57.3)	51 (44.0)
Other	37 (16.5)	13 (7.2)	34 (18.3)	10 (9.7)	6 (5.2)
Taste better	59 (26.3)	62 (34.3)	48 (25.8)	26 (25.2)	47 (40.5)
Dietary restrictions					
Do not follow any dietary restrictions	177 (79.0)	136 (75.1)	148 (79.6)	87 (84.5)	78 (67.2)
Food allergies/lactose intolerance etc	25 (11.2)	15 (8.3)	22 (11.8)	6 (5.8)	12 (10.3)
Halal	6 (2.7)	7 (3.9)	3 (1.6)	1 (1.0)	9 (7.8)
Vegetarian	10 (4.5)	15 (8.3)	10 (5.4)	5 (4.9)	10 (8.6)
Others	6 (2.7)	8 (4.4)	3 (1.6)	4 (3.9)	7 (6.0)
Diet Quality					
Did not meet dietary guidelines of any food group	29 (12.9)	25 (13.8)	23 (12.4)	24 (23.3)	7 (6.0)
Met dietary guidelines of one group	55 (24.6)	46 (25.4)	58 (31.2)	32 (31.1)	11 (9.5)
Met dietary guidelines of two groups	36 (16.1)	24 (13.3)	33 (17.7)	15 (14.6)	12 (10.3)
Met dietary guidelines of three groups	21 (9.4)	28 (15.5)	22 (11.8)	13 (12.6)	14 (12.1)
Met dietary guidelines of four groups	40 (17.9)	22 (12.2)	28 (15.1)	10 (9.7)	24 (20.7)
Met dietary guidelines of all groups	43 (19.2)	36 (19.9)	22 (11.8)	9 (8.7)	48 (41.4)
Exercise					
None	46 (20.5)	37 (20.4)	46 (24.7)	20 (19.4)	17 (14.7)
1–3 times per week	129 (57.6)	115 (63.5)	99 (53.2)	62 (60.2)	83 (71.6)
4–6 times per week	33 (14.7)	20 (11.0)	26 (14.0)	13 (12.6)	14 (12.1)
Every day	16 (7.1)	9 (5.0)	15 (8.1)	8 (7.8)	2 (1.7)
Medical Condition					
No	148 (66.1)	119 (65.7)	117 (62.9)	74 (71.8)	76 (65.5)
Yes	76 (33.9)	62 (34.3)	69 (37.1)	29 (28.2)	40 (34.5)

**Table 3 nutrients-16-03963-t003:** Factors associated with LCSs (low calorie sweeteners) use among women of reproductive age in Australia.

LCS Use Reported	Odds Ratio	95% Confidence Interval	
**Age**			
18–25 year	Reference		
26–35 years	0.64	0.34	1.19
36–49 years	0.42	0.21	0.84
**Ethnicity**			
Caucasians	Reference		
Aboriginal/Torres Strait Islander	1.40	0.42	4.63
Asian/Pacific Islander	0.67	0.24	1.84
Other	1.09	0.40	2.97
South Asian	4.16	1.71	10.08
**Employment**			
Employed	Reference		
A homemaker	0.63	0.32	1.26
Early Retirement, unable to work	0.95	0.42	2.14
**Education**			
Bachelors and bove	Reference		
Certificate or Diploma	0.98	0.54	1.80
Completed year 12	0.73	0.38	1.43
Others	0.88	0.28	2.80
**Marital status**			
Married/de facto relationship	Reference		
Single—never married	0.61	0.36	1.03
Divorsed/Widowed	0.47	0.17	1.34
**Country of birth**			
Australia	Reference		
Other	0.71	0.35	1.44
**BMI** (kg/m^2^)			
18–24	Reference		
25–29	0.22	0.06	0.83
30–34	0.42	0.11	1.68
≥35	0.29	0.07	1.11
**Health Status**			
Excellent	Reference		
Fair	1.18	0.49	2.85
Good	0.79	0.37	1.70
Poor	0.48	0.09	2.75
**Participation in weight management program**			
No	Reference		
Yes	1.92	1.05	3.51
**Reasons for LCS use**			
Health professional advice	Reference		
Diabetes managemnet	1.93	0.50	7.36
Weight loss	5.37	1.86	15.48
Other	1.99	0.58	6.80
Taste better	5.45	1.85	16.10
**Dietary restrcitions**			
No dietary restriction	Reference		
Food allergies/lactose intolerance etc	0.59	0.26	1.34
Halal	1.02	0.24	4.34
Vegetarian	1.61	0.58	4.48
Others	1.03	0.29	3.64
**Excercise**			
None	Reference		
1–3 times per week	1.07	0.60	1.92
4–6 times per week	0.72	0.31	1.69
Every day	0.59	0.20	1.68
**Medical condition**			
No	Reference		
Yes	1.25	0.74	2.09
**Diet quality**			
Met dietary guidelines of all groups	Reference		
Did not meet dietary guildines of any group	1.75	0.77	3.97
Met dietary guidelines of one food group	1.85	0.90	3.79
Met dietary guidelines of two food groups	1.32	0.58	3.00
Met dietary guidelines of three food groups	2.27	0.98	5.27
Met dietary guidelines of four food groups	0.80	0.36	1.76

**Table 4 nutrients-16-03963-t004:** Factors associated with moderate and heavy use of LCSs among women of reproductive age in Australia.

	RRR (95 CI)	RRR (95 CI)
	Class 2 (Moderate LCS Use)	Class 3 (Heavy LCS Use)
**Age**		
18–25 years	Reference	Reference
26–35 years	0.97 (0.44, 2.15)	0.53 (0.25, 0.53)
36–49 years	0.49 (0.2, 1.2)	0.39 (0.17, 0.39)
**Ethnicity**		
Caucasians	Reference	Reference
Aboriginal/Torres Strait Islander	0.83 (0.15, 4.55)	1.38 (0.33, 1.38)
Asian/Pacific Islander	0.25 (0.06, 1.08)	0.64 (0.21, 0.64)
Other	0.98 (0.28, 3.41)	0.45 (0.12, 0.45)
South Asian	0.85 (0.3, 2.43)	0.77 (0.28, 0.77)
**Employment**		
Employed	Reference	Reference
A homemaker	1.45 (0.62, 3.4)	1.29 (0.53, 1.29)
Early Retirement, unable to work	0.33 (0.1, 1.02)	0.38 (0.13, 0.38)
**Education**		
Bachelors and above	Reference	Reference
Certificate or Diploma	0.43 (0.2, 0.91)	0.52 (0.24, 0.52)
Completed year 12	0.41 (0.18, 0.96)	0.57 (0.25, 0.57)
Others	0.21 (0.04, 1)	0.4 (0.08, 0.4)
**Marital Status**		
Married or de facto relationship	Reference	Reference
Single/never married	0.7 (0.36, 1.36)	0.83 (0.43, 0.83)
Divorced/Widowed	0.36 (0.1, 1.31)	0.51 (0.14, 0.51)
**Country of Birth**		
Australia	Reference	Reference
Other	0.68 (0.28, 1.65)	0.89 (0.37, 0.89)
**BMI**		
18–24	Reference	Reference
25–29	1.53 (0.21, 11.1)	1.47 (0.29, 1.47)
30–34	4.8 (0.65, 35.28)	1.98 (0.36, 1.98)
≥35	3.35 (0.47, 23.74)	1.52 (0.3, 1.52)
**Health Status**		
Excellent	Reference	Reference
Fair	1.28 (0.4, 4.08)	1.33 (0.45, 1.33)
Good	0.81 (0.3, 2.23)	0.63 (0.25, 0.63)
Poor	0.92 (0.1, 8.72)	1.1 (0.16, 1.1)
Participation in weight management program		
No	Reference	Reference
Yes	3.78 (1.71, 8.34)	3.94 (1.74, 3.94)
Reasons for using LCS		
Health professional advise	Reference	Reference
Diabetes management	4.27 (0.7, 26.12)	3.0 (0.59, 3)
Weight loss	9.31 (2.18, 39.74)	3.4 (0.88, 3.4)
Other	3.14 (0.62, 15.83)	1.34 (0.27, 1.34)
Taste better	6.13 (1.35, 27.89)	5.19 (1.32, 5.19)
**Exercise**		
No exercise	Reference	Reference
1–3 times per week	1.46 (0.69, 3.11)	1.80 (0.84, 1.8)
4–6 times per week	1.22 (0.42, 3.5)	1.10 (0.37, 1.1)
Every day	1.52 (0.45, 5.18)	0.60 (0.11, 0.6)
**Medical Condition**		
No	Reference	Reference
Yes	1.06 (0.54, 2.05)	1.32 (0.69, 1.32)
**Diet quality**		
Met dietary guidelines of all groups	Reference	
Did not meet dietary guildines of any group	3.28 (1.06, 1.06)	0.21 (0.07, 0.07)
Met dietary guidelines of one food group	1.63 (0.58, 0.58)	0.1 (0.04, 0.04)
Met dietary guidelines of two food groups	1.17 (0.37, 0.37)	0.21 (0.08, 0.08)
Met dietary guidelines of three food groups	1.26 (0.39, 0.39)	0.27 (0.1, 0.1)
Met dietary guidelines of four food groups	0.69 (0.21, 0.21)	0.39 (0.16, 0.16)

Base outcome = Class 1.

## Data Availability

The original contributions presented in the study are included in the article/Appendix A, further inquiries can be directed to the corresponding author.

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
