# Peer review of "Prevalence and Predictors of the Use of Low-Calorie Sweeteners Among Non-Pregnant, Non-Lactating Women of Reproductive Age in Australia"

_nutrients, 2024, doi:10.3390/nu16223963_

Round 1

Reviewer 1 Report

Comments and Suggestions for Authors

An interesting study exploring the prevalence of low-calorie sweetener (LCS) use in women of reproductive age in Australia. However, there are some comments that the authors should respond to:

- Methods. More details should be given on how the FFQ is corrected and obtained for the nutritional groups (are there only 5 items? or are there 5 points on the response scale? -line 89-). On the other hand, the BEVQ-15 should be more detailed. Is it validated in this population?

- Line 112. Was general health self-reported? in paragraph 120-123, not described above?.

- More details should be given about the statistical part, what software was used? what test? What p-value was shown to be significant? How was the model built?

- Results. The self-reported consumption of LCS was 44.7% while the users classified as slightly was 45.9%. Could this indicate that the self-report is not valid? or rather the categorization of the models? On the other hand, Figure 1 is not informative, the data could be incorporated into the main text.

- The p-value of contrast between LCS consumption (yes/no) and LCS classification patterns should be reported. If the variables in Table 2 did not demonstrate significance in the univariate contrasts, they would not need to be included in the model.

- Table 3 and 4 would need to report the p-value.

- Discussion. Paragraphs 235-241, would it be outside the objective of the study?

- Another limitation of the study, which I have already mentioned previously, would be that the LCS categories have been self-reported through consumption frequencies, which can bias the data.

- Conclusions. Linease 287-289, really any nutritional cosmos pattern depends on social, health, or cultural variables, it is not something from LCS. What implications does this data have?

Minor comments:

- Lines 147, put "3".

- Specify RRR acronym from table 4.

Author Response

An interesting study exploring the prevalence of low-calorie sweetener (LCS) use in women of reproductive age in Australia. However, there are some comments that the authors should respond to:

- Methods. More details should be given on how the FFQ is corrected and obtained for the nutritional groups (are there only 5 items? or are there 5 points on the response scale? -line 89-). On the other hand, the BEVQ-15 should be more detailed. Is it validated in this population?

Author's response

More detail about the FFQ is added in the method section

- Line 112. Was general health self-reported? in paragraph 120-123, not described above?.

Author's response

Yes, health status was self-reported and added

- More details should be given about the statistical part, what software was used? what test? What p-value was shown to be significant? How was the model built?

Author's response

We added information on what software was used.

Regarding p-values and statistical significance, we followed the American Statistical Association's Statement for interpreting estimates and confidence intervals instead of statistical significance.

Sources:

Valentin Amrhein, Sander Greenland, Blake McShane, Retire statistical significance. Nature, 2019, vol l567, 305

Wasserstein RL, Lazar NA. The ASA statement on P-values: context, process, and purpose. Am Stat 2016; 70: 129–133.

- Results. The self-reported consumption of LCS was 44.7% while the users classified as slightly was 45.9%. Could this indicate that the self-report is not valid? or rather the categorization of the models? On the other hand, Figure 1 is not informative, the data could be incorporated into the main text.

Author's response

The self-reported data may slightly underrepresent actual use, as it is based on a single question: "Do you consume any artificial sweeteners?" In contrast, the latest class model incorporates a wide range of information to identify distinct latent classes of low-calorie sweetener (LCS) use, based on consumption across various beverages and products. Latent class analysis (LCA) is employed to uncover latent (unobserved) heterogeneity within the sample, allowing for the identification of subgroups within the population that share specific characteristics related to LCS consumption.

 We have now removed Figure 1, only retaining the information in the text.

- The p-value of contrast between LCS consumption (yes/no) and LCS classification patterns should be reported. If the variables in Table 2 did not demonstrate significance in the univariate contrasts, they would not need to be included in the model.

- Table 3 and 4 would need to report the p-value.

Author's response

As mentioned in the above response we followed the American Statistical Association's Statement for interpreting estimates and confidence intervals instead of statistical significance. We a priori identified covariates based on theoretical knowledge, therefore covariates inclusion in the adjusted model was not driven by p-value.

Sources:

Valentin Amrhein, Sander Greenland, Blake McShane, Retire statistical significance. Nature, 2019, vol l567, 305

Wasserstein RL, Lazar NA. The ASA statement on P-values: context, process, and purpose. Am Stat 2016; 70: 129–133.

- Discussion. Paragraphs 235-241, would it be outside the objective of the study?

- Another limitation of the study, which I have already mentioned previously, would be that the LCS categories have been self-reported through consumption frequencies, which can bias the data.

Author's response

We have added one line about the limitation of the LCS method.

We removed that line from the discussion, “These findings suggest that LCS consumption is influenced by a combination of cultural, health-related, and dietary factors”

- Conclusions. Linease 287-289, really any nutritional cosmos pattern depends on social, health, or cultural variables, it is not something from LCS. What implications does this data have?

Author's response

We have modified the conclusion section.  

We removed the sentence “with distinct consumption patterns influenced by cultural, health-related, and socioeconomic factors from the conclusion.”

Reviewer 2 Report

Comments and Suggestions for Authors

It is an interesting work where authors investigated prevalence of LCS and its determinants among Australian women. However, some technical issues should be addressed further if a revision will be invited as following:

1.      It is unclear about LCA application for classification of LCS. Authors used frequency of times per week for each beverage or amount when conducting LCA? How about jam/jellies, candies, and condiments?

2.      For table 4, why did authors not analyze variables about “meeting the dietary guidelines”?

3.      For table 4, why did authors present RRR instead of OR?

4.      A key point should be mentioned about definition of low-calorie sweeteners. In fact, in this work, authors investigated mainly beverage and food products containing sweeteners. how to distinguish these products containing LCS? Maybe authors say something more about this issue in the part of methods.

Author Response

Comments and Suggestions for Authors

It is an interesting work where authors investigated prevalence of LCS and its determinants among Australian women. However, some technical issues should be addressed further if a revision will be invited as following:

  1. It is unclear about LCA application for the classification of LCS. Authors used frequency of times per week for each beverage or amount when conducting LCA? How about jam/jellies, candies, and condiments?

Author's response

We added more information in the method section for further clarity.

  1. For table 4, why did authors not analyze variables about “meeting the dietary guidelines”?

Author's response

We have now added the results of meeting the dietary guidelines and LCS use in table 4.

  1. For table 4, why did authors present RRR instead of OR?

Author's response

The outcome variable in Table 4 was not a binary outcome, instead, the outcome variable had three categories (low, moderate,  and heavy LCS use), therefore, we used multinomial logistic regression and the output of this model is relative risk ratio.

  1. A key point should be mentioned about definition of low-calorie sweeteners. In fact, in this work, authors investigated mainly beverage and food products containing sweeteners. how to distinguish these products containing LCS? Maybe authors say something more about this issue in the part of methods.

Author's response

We have now added the definition of LCS in the method section.

Round 2

Reviewer 1 Report

Comments and Suggestions for Authors

Thanks for this revised version. However, the comments were not deeply detailed. It is not clear how the FFQ corrections are applied, nor the statistical methods used. On the other hand, reporting the p-value is not incompatible with the CI. The explanation for the % in the self-report is also not clear.

Author Response

Reviewers comments  (R2)

Thanks for this revised version. However, the comments were not deeply detailed. It is not clear how the FFQ corrections are applied, nor the statistical methods used. On the other hand, reporting the p-value is not incompatible with the CI. The explanation for the % in the self-report is also not clear.

Author’s response

We did not use an extensive food frequency questionnaire (FFQ), as assessing the intake of specific nutrients was not the aim of our study. Instead, we wanted a proxy measure of compliance with the Australian Dietary Guidelines as a covariate, so we used a brief, six-item dietary screening tool (the Australian Short Dietary Screener), which has been validated in older populations.

Respondents were asked to report the number of serves they consumed from each food group daily or on an average week. Examples of a single serve for each food group were provided in the questionnaire, based on the descriptions in the Australian Dietary Guidelines. The six-item tool includes one question per food group, rather than detailed inquiries into individual food items within each group. Consequently, we have added a note in the limitations section regarding potential reporting or measurement bias in our assessment of dietary guideline compliance. Additionally, we updated the methods section to describe the six-item dietary assessment tool.

Since we did not use an extensive FFQ or assess intake of individual food items within each group, no correction for FFQ measurement error was applied in our analyses.

The explanation for the % in the self-report is also not clear.

Author’s response

In this question, it’s unclear what you are referring to. However, if you mean the self-reported LCS data, we previously noted that self-reports might slightly under-represent actual consumption, as they rely on a single question: 'Do you consume any artificial sweeteners?' The latest classification model, however, uses a broader set of information to identify different patterns or 'latent classes' of LCS use. It categorizes usage based on low-calorie sweetener intake across various beverages and products.

On the other hand, reporting the p-value is not incompatible with the CI.

Author’s response

We did not interpret our results based on significance levels or p-values; instead, we reported the effect estimates and confidence intervals.

Reviewer 2 Report

Comments and Suggestions for Authors

Thanks for authors' response and revsion. The following minor issue is suggested to address futher:

multinomial logistic regression seems to only estimate OR, pls check it carefully.

Author Response

Reviewers comment

Thanks for authors' response and revision. The following minor issue is suggested to address futher:

multinomial logistic regression seems to only estimate OR, pls check it carefully

Author’s response

Thank you for your comment.

Multinomial logistic regression is used to model nominal outcome variables with more than two categories. We used the mlogit command in Stata. This model can also provide relative-risk ratios if you specify the rrr option.

For example: mlogit y x1 x2 i.a, rrr (an example output is provided below).

Reference : Stata manual on mlogit — Multinomial (polytomous) logistic regression.

UCLA: Multinomial Logistic Regression | Stata Data Analysis Examples
